# The Impact of Cropland Use Changes on Terrestrial Ecosystem Services Value in Newly Added Cropland Hotspots in China during 2000–2020

**Tianyi Cai** [1] , **Xueyuan Luo** [2], **Liyao Fan** [3], **Jing Han** [3] **and Xinhuan Zhang** [4],*

1    College of Landscape Architecture and Art, Henan Agricultural University, Zhengzhou 450002, China
2    No. 29 Middle School of Zhengzhou, Zhengzhou 450048, China
3    College of Resources and Environmental Sciences, Henan Agricultural University, Zhengzhou 450002, China
4    State Key Laboratory of Desert and Oasis Ecology, Xinjiang Institute of Ecology and Geography, Chinese Academy of Sciences, Urumqi 830011, China
*    Correspondence: zhangxh@ms.xjb.ac.cn; Tel.: +86-991-782-7314

**Abstract:** The assessment of ecosystem services value is the basis for the realization of integrated management of these services. In recent decades, the hotspots of China's newly added cropland have shifted to the Xinjiang oasis areas, where the ecological environment is relatively fragile. However, the impact of changes in cropland use on the terrestrial ecosystem services value (TESV) in Xinjiang, China, has not been studied in depth, and it is related to the sustainability of the dynamic balance between China's cropland and the sustainable management of natural resources in Xinjiang. This study focuses on Xinjiang and employs the benefit transfer method and five phases (2000, 2005, 2010, 2015 and 2020) of high-resolution and finely classified remote sensing monitoring data of land use to evaluate the impact of changes in cropland use on the TESV from 2000 to 2020. The findings suggest the following: (1) The cropland area in Xinjiang grew from 6.5682 million ha in 2000 to 8.9874 million ha in 2020, demonstrating significant expansion, and it has gone through four stages: rapid expansion, steady expansion, rapid expansion and relative stability. (2) A sharp mutual conversion trend is observed between cropland and other types of land use in Xinjiang. Grassland and unused land are the main sources of new cropland, while lost cropland has been mainly converted back into grassland and construction land. (3) During 2000–2020, although the extensive expansion of cropland and conversion of cropland mainly at the expense of ecological land in Xinjiang have significantly enhanced the provision services of the terrestrial ecosystem (539.49 million USD), ecosystem regulation services (−1508.47 million USD), support services (−1084.47 million USD) and cultural services (−565.05 million USD) experienced losses. Consequently, an overall loss in the TESV has ensued. This study provides new insights that help re-examine the sustainability issue of the spatial transfer of cropland in China, and it also offers guidance for the realization of the sustainable management of natural resources in Xinjiang.

**Keywords:** cropland expansion; cropland use conversion; ecosystem services value; land use remote sensing monitoring data; Xinjiang

## 1. Introduction

Ecosystem services profoundly impact the supply of natural resources, human well-being and sustainable development [1]. The assessment of ecosystem services value (ESV) reconstructs the relationship between human beings and nature, thus making it possible to aggregate and compare different ecosystems by monetary means [2]; furthermore, it reminds people to regard natural assets as an important component of inclusive wealth, well-being and sustainability. Ultimately, it is the basis for realizing integrated management of ecosystem services [3]. Land is a mosaic of various ecological systems in a region, and changes in land use are the most direct manifestation of the interaction between humans

and natural ecosystems [4]. Moreover, the impacts of the different types of land use on ESV significantly differ based on land quantity and changes in the spatial pattern of land use [5–9]. In this regard, cultivating cropland is the type of land use most closely related to human survival and development [10,11], and the development and utilization of cropland have significantly influenced the sustainable development of the terrestrial ecosystem [12,13]. Therefore, clarifying the impact of changes in cropland use on the terrestrial ecosystem services value (TESV) is critical for the coordinated guarantee of food security and ecological security, and the realization of the sustainable management of natural resources.

Globally, urbanization and food production sectors pose fierce direct competition on land [14]. Although urban expansion already occupies a substantial area of cropland worldwide [15], population growth and dietary shifts continue to drive global cropland expansion [16]. In the past 300 years, the global cropland area has expanded by about five times [12]. In the last 70 years, the frontier of global agricultural expansion has shifted from Europe and North America to the tropics, with new cropland being established largely at the expense of forests and grasslands [12,17–19]. During 1992–2015, changes in global cropland were responsible for an absolute loss of 166.82 billion USD, equivalent to 1.17% of the global TESV in 1992 [8]. The impact of changes in cropland on the TESV was most significant in South America and Africa, but it was not obvious in Oceania, Asia and Europe [8].

As China is the most populous country in the world, its cropland utilization holds important strategic significance for its own and global food security. Since China's reform and opening-up, its rapid urbanization and industrialization process has led to the conversion of several high-quality cropland resources into non-agricultural construction land [20]. However, the total area of cropland in China has remained relatively steady since 2000. This is because while cropland has shrunk in southern China, it has continued to expand in northern China, and the focus of cropland reclamation in the north has shifted to the oasis agricultural area in Xinjiang [21–25]. Ultimately, this spatial transfer of cropland distribution has maintained the dynamic balance of the total amount of cropland in China. However, Xinjiang—which is a hotspot for newly added cropland in China—is located in the inland northwest of China and constitutes a major part of China's arid area, with scarce water resources and an extremely fragile ecological environment [26]. Therefore, some scholars have expressed concerns about the sustainability as well as ecological and environmental risks of this unusual phenomenon. In this regard, the existing literature mainly refers to the impact of the spatial transfer of China's cropland on the spatial mismatch between food production and cropland resources [27]; the natural suitability of cropland [25]; the quality gap in cropland [28] and the change in potential agricultural productivity [29]; and wind erosion, irrigation water consumption, fertilizer use and natural habitats in newly developed areas of cropland [23]. However, the impact of China's cropland spatial transfer on the TESV in hotspots for newly added cropland has not yet been comprehensively studied.

Many scholars have focused on the spatiotemporal dynamics, transformation mode and driving mechanism of long-term changes in cropland in Xinjiang from the perspective of changes in land use based on remote sensing monitoring data of land use [30–34]; however, they did not perform quantitative analyses of the impacts of changes in cropland use on ecological environment. Some scholars have conducted empirical studies on Xinjiang [35], Kashgar region [36], Manas River Basin [37], Ebinur Lake Wetland National Nature Reserve [38] and the northern slope economic belt of the Tianshan Mountains [39] from the perspective of changes in land use and the ESV response, providing useful references for the sustainable management of land ecosystem services in different regions of Xinjiang. However, cropland is only considered as one of the types of land use in these studies, thus offering insufficient pertinence. Additionally, the research on individual typical regions in Xinjiang cannot address the question regarding the overall impact of the transfer of China's newly added cropland to the Xinjiang oasis area on its TESV.

In terms of research methods, the main methods to estimate ESV include the benefit transfer method, market price method, productivity method and travel cost method [40]. The benefit transfer method facilitates decision-making when estimating ESV at broad geographical scales, as it offers quick assessment and low-cost primary data collection [7,41–44]. This method involves estimating the economic benefits gathered from one site and applying them to another [7]. Specifically, this method assesses the ESV for a study area using monetary values pre-assigned to each biome or land-use/land-cover change (LUCC) type [7]. In 1997, Costanza et al. [2] first proposed this method and assessed the service value of 17 ecosystem services in 16 biomes around the world. However, the method proposed by Costanza et al. is not completely suitable for application in China, for example, as it underestimates the ESV of cropland and fails to reflect the actual economic level of developing countries [45]. Chinese scholars Xie et al. [45] reclassified China's mainland into six ecosystem types, four service function types and nine service subfunctions based on Costanza et al.'s work and China's ecological characteristics. They obtained the Chinese ESV equivalent table based on the questionnaire-based survey of Chinese ecologists, which greatly promoted ESV evaluation in China. However, Xie et al. also highlighted that China has a vast territory and is characterized by significant spatial heterogeneity in the strength and type of its ecosystem service functions. Only by improving the ESV equivalent table according to local conditions can the accuracy of ESV assessment in local regions be improved [46]. Moreover, the research period of the existing literature on the change in cropland use in Xinjiang does not extend beyond 2015. With the passage of time and continuous updating of data, it is necessary to improve the timeliness of the research and provide managers with a decision-making basis that considers the change process and recent status. Additionally, the land cover classification and spatial resolution of the remote sensing monitoring data of land use, on which the assessment of ESV depends, are also important to the accuracy of the assessment results [47].

Based on the above-mentioned research basis and gaps, this paper takes Xinjiang, a hotspot for newly added cropland in China, as the research area. Based on the 30 m resolution remote sensing monitoring dataset of land use that includes six primary and 25 secondary types of land use and is issued by the Resource and Environment Data Center of the Chinese Academy of Sciences, this study employs the benefit transfer method and the improved ESV equivalent table to evaluate the impact of changes in cropland use on the TESV and its spatial difference characteristics in Xinjiang from 2000 to 2020. This paper aims to provide new insights from the perspective of ecosystem services to re-examine the profound impact of China's cropland spatial transfer and its dynamic balance, and it also hopes to offer guidance for the realization of the sustainable management of natural resources in Xinjiang.

## 2. Materials and Methods

### 2.1. Study Area

Xinjiang, short for the Xinjiang Uygur Autonomous Region, 73°40′–96°23′ E and 34°25′–49°10′ N, is located in the middle of Eurasia and in the northwest of China, and it has a typical temperate continental arid climate. It is the largest provincial administrative region in China and also constitutes the principal part of the arid region in northwest China [26]. It has three mountain ranges, namely the Altai Mountains, Tianshan Mountains and Kunlun Mountains, as well as two inland basins—the Junggar Basin and the Tarim Basin. They form a topography and landform of "three mountain ranges surrounding two basins" and an ecological landscape pattern of "mountain–oasis–desert", as shown in Figure 1; among them, cropland is the main body of Xinjiang's artificial oases and is integral to supporting the oasis stability and regional ecological security in Xinjiang [11].

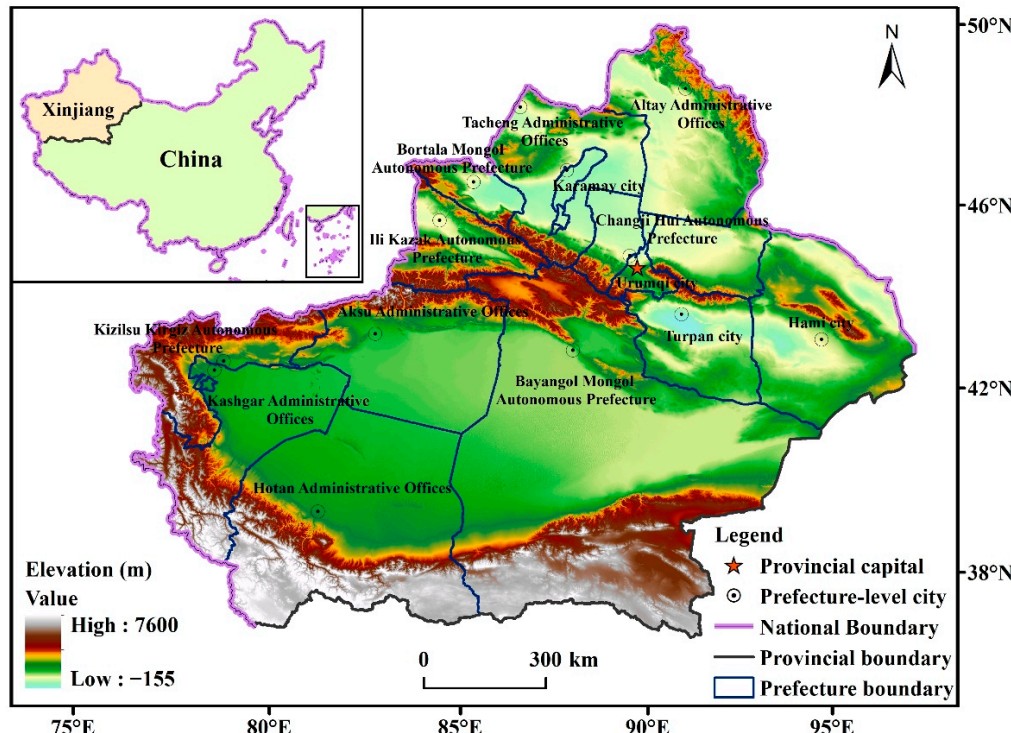

**Figure 1.** The geographic location of Xinjiang and its administrative division.

Since the 21st Century, a significant spatial shift has occurred in the distribution of cropland in China. The concentration of newly added cropland has shifted from northeast China to the oasis area in Xinjiang, where the ecological environment is relatively fragile [21–24]. As of 2020, Xinjiang had 8.99 million ha of cropland, accounting for about 5% of the total land area of Xinjiang, and it had a total population of 25.90 million people, with an urbanization rate of 56.53% [48]. In Xinjiang, nearly half of the population lives in rural areas, with almost 72% of the total rural employees engaged in agriculture [48]. Farmers' livelihoods are highly dependent on agriculture and cropland [11].

According to the Xinjiang Statistical Yearbook data [48], in 2020, there were 14 prefecture-level administrative divisions in Xinjiang (Figure 1): Urumqi city (UMQC), Karamay city (KRMC), Turpan city (TUPC), Hami city (HAMC), Changji Hui Autonomous Prefecture (CHAP), Ili Kazak Autonomous Prefecture (IKAP), Tacheng Administrative Offices (TCAO), Altay Administrative Offices (ATAO), Bortala Mongol Autonomous Prefecture (BTAP), Bayangol Mongol Autonomous Prefecture (BYAP), Aksu Administrative Offices (ASAO), Kizilsu Kirgiz Autonomous Prefecture (KKAP), Kashgar Administrative Offices (KSAO) and Hotan Administrative Offices (HTAO). Additionally, there were some county-level cities under the jurisdiction of the Xinjiang Production and Construction Corps within the administrative region of Xinjiang. In this study, to reveal the response characteristics of the TESV pursuant to changes in cropland use from the perspective of the whole territory of Xinjiang, we incorporated four county-level cities (Shihezi City, Wujiaqu City, Alar City and Tumushuk City) under the jurisdiction of Xinjiang Production and Construction Corps into TCAO, CHAP, ASAO and KSAO, respectively.

## 2.2. Methods

### 2.2.1. Method of Analysis of Cropland Quantity Change

The net change area of cropland (NCAC) and the annual change rate of cropland (ACRC) can, respectively, depict the change characteristics of the scale and speed of cropland use in a certain region during a certain period. These two indexes have been widely used in recent analyses of cropland quantity change [22,30,34]. Similarly, this study also

employs the two indexes to describe the characteristics of cropland quantity change in the study area.

The NCAC is calculated as follows:

$$A_{i,t} = A_{i,t2} - A_{i,t1} \tag{1}$$

where $A_{i,t1}$ and $A_{i,t2}$ represent the area of cropland in the $i$ study area at the beginning and the end of the study period, from $t1$ to $t2$, respectively. When the value of $A_{i,t}$ is positive, the larger the value, the larger the scale of cropland expansion during the period; when it is negative, the opposite is true.

The ACRC is calculated as follows:

$$V_{i,t} = (A_{i,t2} - A_{i,t1}) / A_{i,t1} \times (1/t) \times 100\% \tag{2}$$

In this formula, when $V_{i,t}$ is greater than 0, it indicates that the cropland in the area is increasing, and when it is less than 0, it indicates that the cropland in the area is decreasing. When $V_{i,t}$ is greater than 0, the larger the value, the faster the expansion of cropland.

### 2.2.2. Method of Analysis of Cropland Use Conversion

The Markov land use transition matrix can accurately reveal the conversion process between cropland and other land use categories, as well as the area and spatial location of the conversion, and it has been widely used in recent land use change and simulation analyses [21,30,41,43]. Similarly, this study employs this method to describe the characteristics of cropland use transition in the study area. The calculation formula is as follows:

$$P_{gain(i),j} = (P_{i,j} - P_{j,i}) / (P_{i.} - P_{.i}) \times 100, i \neq j \tag{3}$$

$$P_{loss(i),j} = (P_{j,i} - P_{i,j}) / (P_{i.} - P_{.i}) \times 100, i \neq j \tag{4}$$

where $P_{gain(i),j}$ is the proportion of land use type $i$ converted to land use type $j$ in the net increased area of all the land use types in the $i$th row of the transition matrix, that is, the contribution rate of the conversion. $P_{loss(i),j}$ is the proportion of land use type $i$ converted to land use type $j$ in the net decreased area of all the land use types in the $i$th row of the transition matrix. $P_{i,j}$ and $P_{j,i}$ are single values in the transition matrix table. $P_{i.}$ is the area of a land use type of the $i$th row at the end of the period, and $P_{.i}$ is the area of the land use type at the beginning of the period.

### 2.2.3. Method of ESV Estimation

- Calculation of unit value of ecosystem services

In 1997, Costanza et al. [2] proposed the principles and methods to evaluate 17 ecosystem services in 16 biomes around the world, which clarified the estimation of ESV in a scientific sense. Based on Costanza et al.'s work foundation and its shortcomings, in combination with China's ecological characteristics, Chinese scholars Xie et al. [45] classified China's mainland into six types of ecosystems (cropland, woodland, grassland, wetland, river/lake and desert) and four types of service functions (provision services, regulation services, support services and cultural services). Then, Xie et al. conducted a questionnaire-based survey of 700 Chinese ecologists and summarized an equivalent factor of ESV per unit area suitable for an evaluation of ESV at the Chinese scale (Table 1) [45]. According to Xie et al. [45], natural grain output value per unit area of cropland is calculated as follows:

$$E_a = (1/7) \times \sum_{i=1}^{n} (m_i p_i q_i / M)(i = 1, 2, \dots, n) \tag{5}$$

In this formula, $E_a$ represents the natural economic value of grain production service provided per unit sown area of farmland (Yuan/ha); $i$ is the grain species; $p_i$ is the average grain price (Yuan/ton); $q_i$ refers to the yield of grain per unit area of $i$ (ton/ha); $m_i$ refers

to the sown area of *i* (ha); and *M* refers to the total area of grain crops (ha). Additionally, 1/7 represents the ratio of the economic value provided by the natural ecosystem without human input to the economic value of grain production service provided by the per unit cropland area [49].

**Table 1.** Equivalent value per unit area of ecosystem services in China.

| Service Type Categories | Service Type Subcategories | Cropland | Forest | Grassland | River/Lake | Wetland | Desert |
|---|---|---|---|---|---|---|---|
| Provision services | FP | 1.00 | 0.33 | 0.43 | 0.53 | 0.36 | 0.02 |
| | RMP | 0.39 | 2.98 | 0.36 | 0.35 | 0.24 | 0.04 |
| Regulatory services | GR | 0.72 | 4.32 | 1.50 | 0.51 | 2.41 | 0.06 |
| | CR | 0.97 | 4.07 | 1.56 | 2.06 | 13.55 | 0.13 |
| | HR | 0.77 | 4.09 | 1.52 | 18.77 | 13.44 | 0.07 |
| | WD | 1.39 | 1.72 | 1.32 | 14.85 | 14.40 | 0.26 |
| Support services | SC | 1.47 | 4.02 | 2.24 | 0.41 | 1.99 | 0.17 |
| | BP | 1.02 | 4.51 | 1.87 | 3.43 | 3.69 | 0.40 |
| Cultural services | PAL | 0.17 | 2.08 | 0.87 | 4.44 | 4.69 | 0.24 |
| | Total | 7.90 | 28.12 | 11.67 | 45.35 | 54.77 | 1.39 |

Notes: Provision services—Food production (FP) and Raw material production (RMP); Regulatory services—Gas regulation (GR), Climate regulation (CR), Hydrological regulation (HR) and Waste decomposition (WD); Support services–Soil conservation (SC) and Biodiversity protection (BP); and Cultural services—Provide aesthetic landscape (PAL).

　　Due to its vast territory, China has significant regional differences in the types of its food crops, the production capacity of cropland and the market price level of grain. Therefore, this study relies on the calculation method of Xie et al. [45] Equation (5) is used to calculate the economic value of single cropland ecosystem service in Xinjiang combined with the actual grain production and price level in Xinjiang. According to the Xinjiang Statistical Yearbook [48], the main food crops in Xinjiang are wheat and maize; in this regard, from 2000 to 2020, the annual average sown areas of wheat and maize were 935.74 thousand ha and 739.00 thousand ha, respectively, and the annual average yields of wheat and maize per unit sown area were 5.43 ton/ha and 8.48 ton/ha, respectively. According to the China Yearbook of Agricultural Price Survey [50], in 2020, the market prices of wheat and maize in Xinjiang were 2.66 yuan/kg and 2.24 yuan/kg, respectively. Moreover, in 2020, the average exchange rate of USD to RMB was 6.8974. Based on the above-mentioned data and Equation (5), the ESV of Xinjiang was calculated as 340.73 USD/ha·year. Then, we multiplied this benchmark unit price with the ESV equivalent table (Table 1) to obtain the per unit ESV supplied by the ecosystem after correction in Xinjiang (Table 2).

**Table 2.** ESV per unit area of ecosystem after correction in Xinjiang (unit: USD/ha) (2020 prices).

| Service Type Categories | Service Type Subcategories | Cropland | Forest | Grassland | River/Lake | Wetland | Desert |
|---|---|---|---|---|---|---|---|
| Provision services | FP | 340.73 | 112.44 | 146.51 | 180.59 | 122.66 | 6.81 |
| | RMP | 132.88 | 1015.38 | 122.66 | 119.26 | 81.78 | 13.63 |
| Regulatory services | GR | 245.33 | 1471.95 | 511.10 | 173.77 | 821.16 | 20.44 |
| | CR | 330.51 | 1386.77 | 531.54 | 701.90 | 4616.89 | 44.29 |
| | HR | 262.36 | 1393.59 | 517.91 | 6395.50 | 4579.41 | 23.85 |
| | WD | 473.61 | 586.06 | 449.76 | 5059.84 | 4906.51 | 88.59 |
| Support services | SC | 500.87 | 1369.73 | 763.24 | 139.70 | 678.05 | 57.92 |
| | BP | 347.54 | 1536.69 | 637.17 | 1168.70 | 1257.29 | 136.29 |
| Cultural services | PAL | 57.92 | 708.72 | 296.44 | 1512.84 | 1598.02 | 81.78 |
| | Total | 2691.77 | 9581.33 | 3976.32 | 15,452.11 | 18,661.78 | 473.61 |

- Calculation of ESV changes caused by cropland use conversion

As natural ecosystems rely on land as a carrier, changes in land use may greatly affect the ESV. Therefore, when assessing changes in ESV due to land use conversion, an agency method is widely used, that is, matching the type of land use with the equivalent biological community [7]. Using Song and Deng's [7] method, this study established the correspondence between six types of ecosystems and twenty-five secondary types of land use (Table 3). Additionally, we set the coefficient of ecosystem supply, regulation, support and cultural services value of built-up land as 0, according to the research of Costanza et al. [2].

**Table 3.** The corresponding relationship between ecosystem and land use types.

| Ecosystem Type | Land Use Secondary Classification of Chinese Academy of Sciences |
|---|---|
| Cropland ecosystem | 11 paddy field; 12 dry land |
| Woodland ecosystem | 21 forest land; 22 shrub land; 23 sparse woodland; 24 other woodland |
| Grassland ecosystem | 31 high coverage grassland; 32 medium coverage grassland; 33 low coverage grassland |
| Wetland ecosystem | 44 glacier and firn; 46 beach land; 64 marshland |
| River/lake ecosystem | 41 river and canal; 42 lake; 43 water reservoir, pit and pod |
| Desert ecosystem | 61 sandy land; 62 gobi land; 63 saline and alkaline land; 65 bare land; 66 bare rock land; 67 others unused land |

ESV change caused by cropland use conversion is calculated as follows:

$$\Delta ESV = vc_j \times \Delta s_j - vc_i \times \Delta s_i \qquad (6)$$

where $\Delta ESV$ represents the net change of ESV caused by cropland use conversion (in and out); $vc_j$ is the ESV per unit area of $j$th other land use types that were converted to cropland; $vc_i$ is the ESV per unit area of $i$th other land use types that were converted from cropland; $\Delta s_j$ and $\Delta s_i$ are the areas of other land types converted to cropland and cropland converted to other land types, respectively.

*2.3. Data Sources*

The data regarding the types of land use observed in Xinjiang during the five phases of 2000, 2005, 2010, 2015 and 2020 (Figure 2) were obtained from the remote sensing monitoring dataset of land use issued by the Resource and Environment Science and Data Center of the Chinese Academy of Sciences (http://www.resdc.cn, accessed on 10 May 2022); the dataset was at a spatial resolution of 30 m. The data sources from 2000, 2005 and 2010 were mainly Landsat Thematic Mapper (TM) remote sensing images of each phase, and the data sources from 2015 and 2020 were mainly Landsat 8 Operational Land Imager (OLI) remote sensing images of each phase. Finally, the product data were generated through manual visual interpretation. The types of land use include six primary types (cropland, woodland, grassland, water body, built-up land and unused land) and twenty-five secondary types (Table 3). The accuracy of the six classes of land use was above 94.3%, and the overall accuracy of the twenty-five subclasses was above 91.2% [21,22,24]. Administrative boundary vector data were obtained from the basic geographic databases of China at a 1:1 million scale released by the China National Catalog Service for Geographic Information (http://www.webmap.cn, accessed on 5 January 2022). The sown area and output of grain crops in Xinjiang were derived from the Xinjiang Statistical Yearbook, and the market prices of wheat and corn were derived from the China Yearbook of Agricultural Price Survey.

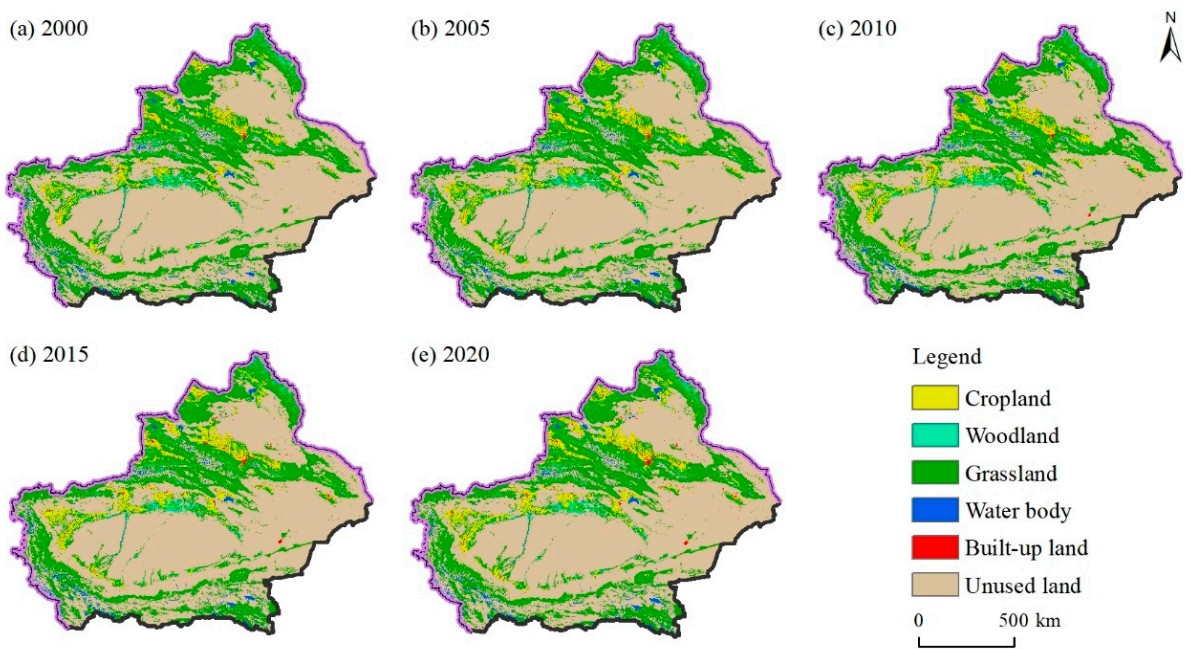

**Figure 2.** Land use types and spatial distribution in Xinjiang: (**a**) land use in 2000; (**b**) land use in 2005; (**c**) land use in 2010; (**d**) land use in 2015; and (**e**) land use in 2020.

## 3. Results

### 3.1. Change Characteristics of Cropland Quantity

#### 3.1.1. Overall Characteristics

From 2000 to 2020, the cropland area in Xinjiang continued to increase from 6.5682 million ha to 8.9874 million ha, with an average annual expansion rate of 1.84%, and it underwent four stages: rapid expansion (the ACRC was about 2.76% during 2000–2005), steady expansion (the ACRC was about 1.58% during 2005–2010), rapid expansion (the ACRC was about 2.23% during 2010–2015) and relative stability (the ACRC was about 0.06% during 2015–2020) (Table 4). In general, the cropland area in Xinjiang underwent significant expansion during 2000–2015, while it remained stable during 2015–2020.

**Table 4.** Change characteristics of cropland quantity in Xinjiang during 2000–2020.

| Year | Area of Cropland ($10^4$ ha) | Period | NCAC ($10^4$ ha) | ACRC (%/yr) |
|------|------------------------------|--------|------------------|-------------|
| 2000 | 656.82 | 2000–2005 | 90.54 | 2.76 |
| 2005 | 747.36 | 2005–2010 | 58.98 | 1.58 |
| 2010 | 806.35 | 2010–2015 | 89.80 | 2.23 |
| 2015 | 896.15 | 2015–2020 | 2.59 | 0.06 |
| 2020 | 898.74 | 2000–2020 | 241.92 | 1.84 |

#### 3.1.2. Regional Differences

In terms of different prefectures in Xinjiang, obvious spatial heterogeneity was observed in the changes in cropland quantity (Table 5 and Figure 3). In terms of the overall study period (2000–2020), the cropland quantity of all prefectures in Xinjiang displayed a trend of expansion. Among them, the NCAC in ASAO was the largest (0.4479 million ha), while the NCAC in UMQC was the smallest (3500 ha). The top five prefectures (ASAO, TCAO, KSAO, BYAP and ATAO) in terms of cropland expansion area contributed 71.18% of the total newly added cropland area in Xinjiang, indicating that cropland expansion in Xinjiang features significant spatial concentration distribution. In terms of the ACRC, KRMC was the fastest (11.32%/yr); additionally, ATAO (3.51%/yr), BTAP (2.77%/yr), BYAP (2.67%/yr), TCAO (2.47%/yr) and ASAO (2.11%/yr) were also faster than the average expansion rate in Xinjiang during 2000–2020.

**Table 5.** Differences in the NCAC and ACRC in different prefectures of Xinjiang during 2000–2020.

| Name | 2000–2005 | | 2005–2010 | | 2010–2015 | | 2015–2020 | | 2000–2020 | |
|---|---|---|---|---|---|---|---|---|---|---|
| | NCAC (10⁴ ha) | ACRC (%) | NCAC (10⁴ ha) | ACRC (%) | NCAC (10⁴ ha) | ACRC (%) | NCAC (10⁴ ha) | ACRC (%) | NCAC (10⁴ ha) | ACRC (%) |
| ASAO | 10.58 | 1.99 | 11.54 | 1.97 | 22.51 | 3.50 | 0.16 | 0.02 | 44.79 | 2.11 |
| KSAO | 4.55 | 0.87 | 10.36 | 1.91 | 19.96 | 3.35 | 0.07 | 0.01 | 34.93 | 1.68 |
| TCAO | 27.56 | 6.30 | 6.94 | 1.21 | 8.56 | 1.40 | 0.15 | 0.02 | 43.21 | 2.47 |
| CHAP | 6.74 | 1.52 | 6.27 | 1.31 | 5.15 | 1.01 | −0.18 | −0.03 | 17.98 | 1.01 |
| IKAP | 8.07 | 1.93 | 2.53 | 0.55 | 3.76 | 0.80 | −0.20 | −0.04 | 14.15 | 0.85 |
| BYAP | 6.47 | 2.56 | 7.84 | 2.75 | 12.66 | 3.90 | 0.01 | 0.00 | 26.97 | 2.67 |
| ATAO | 8.65 | 5.45 | 3.75 | 1.86 | 7.40 | 3.35 | 2.49 | 0.96 | 22.30 | 3.51 |
| HTAO | 2.12 | 1.28 | 2.14 | 1.21 | 5.07 | 2.71 | 0.44 | 0.21 | 9.76 | 1.47 |
| BTAP | 5.97 | 5.81 | 2.16 | 1.62 | 3.33 | 2.32 | −0.07 | −0.05 | 11.38 | 2.77 |
| HAMC | 2.36 | 3.63 | 1.87 | 2.44 | 0.23 | 0.26 | −0.01 | −0.01 | 4.45 | 1.71 |
| KKAP | 0.86 | 1.64 | 0.42 | 0.74 | 1.75 | 2.99 | −0.01 | −0.01 | 3.02 | 1.45 |
| TUPC | 0.53 | 0.86 | 0.39 | 0.61 | −0.31 | −0.46 | −0.00 | −0.01 | 0.62 | 0.25 |
| KRMC | 5.30 | 29.92 | 1.75 | 3.97 | 0.98 | 1.85 | −0.02 | −0.03 | 8.02 | 11.32 |
| UMQC | 0.78 | 1.44 | 1.03 | 1.76 | −1.23 | −1.94 | −0.22 | −0.39 | 0.35 | 0.16 |

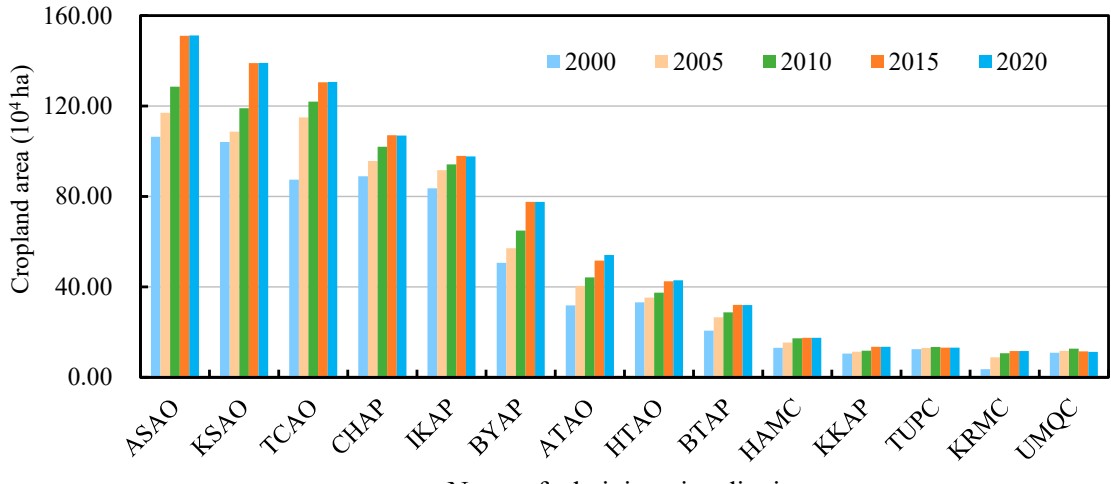

**Figure 3.** Changes in cropland area in different prefectures of Xinjiang during 2000–2020.

In terms of the different study periods, stage differences were also found in the changes in the cropland scale of various prefectures in Xinjiang (Table 5). From 2000 to 2005, the newly added cropland in Xinjiang was mainly distributed in TCAO, ASAO, ATAO and IKAP, accounting for 60.59% of the total newly added cropland in Xinjiang. From 2005 to 2010, the newly added cropland in Xinjiang was mainly distributed in ASAO, KSAO, BYAP, TCAO and CHAP, accounting for 72.82% of the total newly added cropland in Xinjiang. From 2010 to 2015, the cropland scale of UMQC (the ACRC was −1.94%/yr) and TUPC (the ACRC was −0.46%/yr) began to shrink, while the scale of cropland in other prefectures continued to expand, and newly added cropland in Xinjiang was mainly distributed in ASAO, KSAO, BYAP, TCAO and ATAO, accounting for 79.16% of the total newly added cropland. From 2015 to 2020, the scale of cropland in six prefectures (ATAO, HTAO, ASAO, TCAO, KSAO and BYAP) in Xinjiang demonstrated a slight increase, while the scale of cropland in the remaining eight prefectures was shrinking.

### 3.2. Process of Cropland Use Conversion

#### 3.2.1. Sources of the Newly Added Cropland

Table 6 reports the sources of new cropland in Xinjiang during 2000–2020. It shows that the percentages of grassland converted into new cropland during the four periods were

89.43%, 87.25%, 78.70% and 30.08%, respectively. The percentages of unused land converted into new cropland during the four periods measured 9.15%, 11.68%, 17.50% and 53.62%, respectively. The proportion of woodland, water bodies and built-up land converted to cropland was relatively small. Generally, grassland was the main source used for cropland expansion in Xinjiang during the study period. This demonstrates that cropland in Xinjiang was expanded at the expense of large amounts of ecological land. However, with the strengthening of the ecological protection of grassland and the implementation of the policy of returning farmland to grassland, the proportion of new cropland converted from grassland gradually decreased. Conversely, with the progress in the development of water and soil resources and the utilization of technology, the proportion of reclaimed unused land in the newly added cropland continuously increased.

**Table 6.** Statistics on the sources of newly added cropland in Xinjiang during 2000–2020.

| Period | Woodland (WL) | | Grassland (GL) | | Water Body (WB) | | Built-Up Land (BL) | | Unused Land (UL) | |
|---|---|---|---|---|---|---|---|---|---|---|
| | Area ($10^3$ ha) | Percent (%) | Area ($10^3$ ha) | Percent (%) | Area ($10^3$ ha) | Percent (%) | Area ($10^3$ ha) | Percent (%) | Area ($10^3$ ha) | Percent (%) |
| 2000–2005 | 5.36 | 0.56 | 848.36 | 89.43 | 4.54 | 0.48 | 3.61 | 0.38 | 86.77 | 9.15 |
| 2005–2010 | 3.97 | 0.66 | 523.80 | 87.25 | 2.49 | 0.42 | 0.00 | 0.00 | 70.10 | 11.68 |
| 2010–2015 | 29.15 | 2.70 | 849.25 | 78.70 | 4.64 | 0.43 | 7.19 | 0.67 | 188.83 | 17.50 |
| 2015–2020 | 3.59 | 4.44 | 24.29 | 30.08 | 2.26 | 2.80 | 7.32 | 9.07 | 43.31 | 53.62 |

Judging from the spatial distribution of the two main sources of newly added cropland (Figure 4), significant differences in time and prefectures were noted. From 2000 to 2005 (Figure 4a), the grassland occupied by cropland was mainly distributed in TCAO (28.29%), IKAP (12.43%) and ATAO (9.73%) in the north of Xinjiang, while the unused land occupied by cropland was mainly concentrated in TCAO (37.77%), BYAP (19.22%) and ASAO (12.06%). From 2005 to 2010 (Figure 4b), the main distribution areas of cropland cultivated on grassland shifted to ASAO (19.54%), KSAO (19.00%) and BYAP (12.53%) in southern Xinjiang, while the unused land occupied by cropland was mainly in ASAO (19.40%), KSAO (18.23%) and BYAP (12.99%) in southern Xinjiang. From 2010 to 2015 (Figure 4c), cropland established on grassland was mainly distributed in KSAO (22.89%), ASAO (20.86%) and BYAP (10.52%) in southern Xinjiang, while cropland cultivated on unused land was mainly concentrated in BYAP (23.37%), ATAO (20.22%) and ASAO (20.06%). From 2015 to 2020 (Figure 4d), due to the rigid constraints of the "three red lines of water resources" management, the scale of newly added cropland in Xinjiang was extremely small. During this period, unused land was the main source of new cropland, which was mainly concentrated in ATAO (57.95%), which is characterized by relatively rich water resources.

### 3.2.2. Destinations of Lost Cropland

Table 7 reports the destinations of lost cropland in Xinjiang during 2000–2020. From the table, it can be observed that the highest contribution rate of the cropland was noted in the case of converting cropland into grassland during the four periods, measuring 61.82%, 49.94%, 56.91% and 40.38%, respectively. The second highest contribution rates were observed in the case of the occupation of cropland by construction land in the process of urbanization and industrialization during the four periods, measuring 30.64%, 42.73%, 37.02% and 37.70%, respectively. The proportions of cropland converted to woodland, water bodies and unused land were relatively small. In general, with the strengthening of ecological environmental protection in China, ecological conversion (cropland conversion into grassland) under human intervention was the main direction of the reduction of cropland in Xinjiang. With the continuous advancement of urbanization and industrialization, the occupation of cropland by urban and rural construction space has become another major factor causing the loss of cropland in Xinjiang.

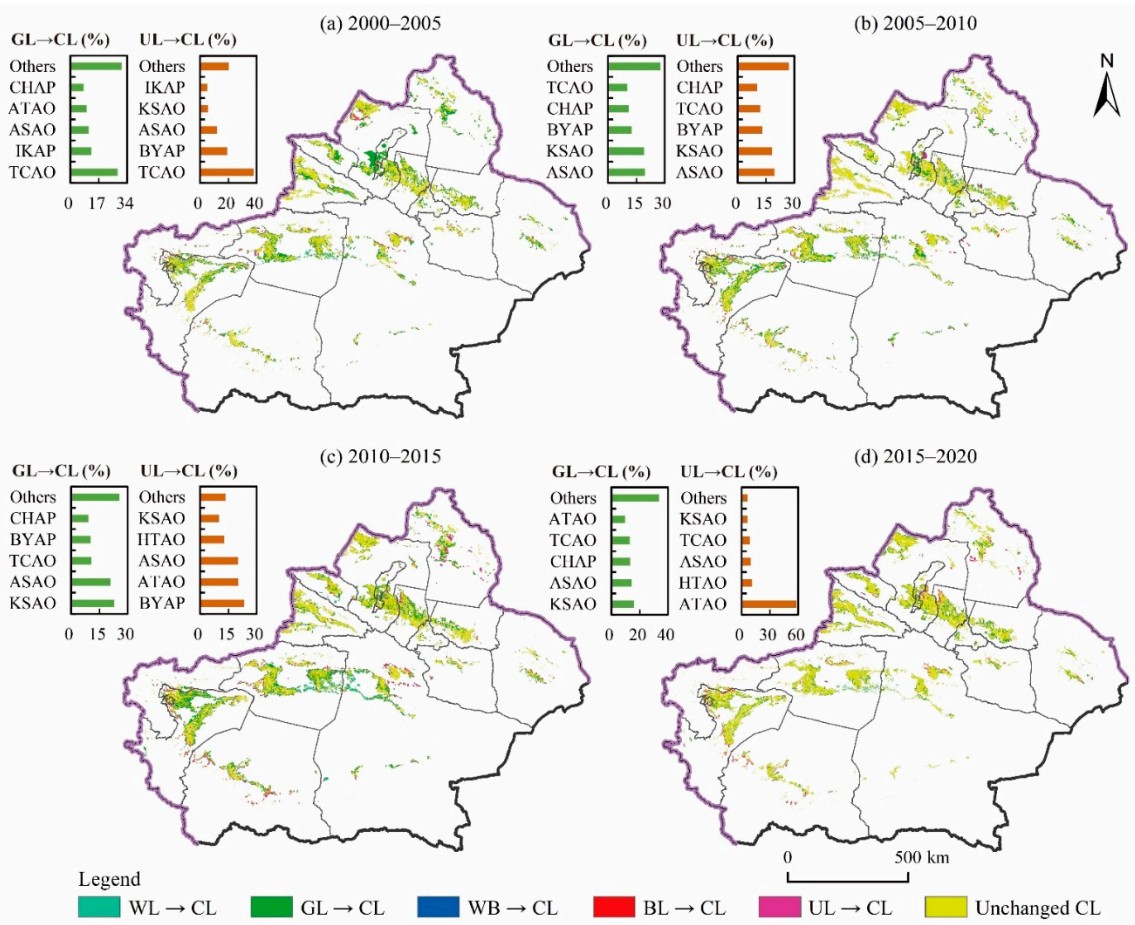

**Figure 4.** Spatial distribution of newly added cropland in Xinjiang: (**a**) during 2000–2005; (**b**) during 2005–2010; (**c**) during 2010–2015; and (**d**) during 2015–2020.

**Table 7.** Statistics on the destinations of lost cropland in Xinjiang during 2000–2020.

| Period | Woodland (WL) | | Grassland (GL) | | Water Body (WB) | | Built-Up Land (BL) | | Unused Land (UL) | |
|---|---|---|---|---|---|---|---|---|---|---|
| | Area (10³ ha) | Percent (%) | Area (10³ ha) | Percent (%) | Area (10³ ha) | Percent (%) | Area (10³ ha) | Percent (%) | Area (10³ ha) | Percent (%) |
| 2000–2005 | 0.68 | 1.58 | 26.67 | 61.82 | 1.53 | 3.54 | 13.22 | 30.64 | 1.04 | 2.42 |
| 2005–2010 | 0.21 | 1.99 | 5.26 | 49.94 | 0.37 | 3.55 | 4.50 | 42.73 | 0.19 | 1.79 |
| 2010–2015 | 4.46 | 2.46 | 103.02 | 56.91 | 1.12 | 0.62 | 67.02 | 37.02 | 5.41 | 2.99 |
| 2015–2020 | 3.25 | 5.90 | 22.25 | 40.38 | 2.99 | 5.43 | 20.77 | 37.70 | 5.83 | 10.59 |

Judging from the spatial distribution of the two main destinations of the lost cropland (Figure 5), there are also significant differences between time and prefectures. From 2000 to 2005 (Figure 5a), the conversion of cropland into grassland was mainly distributed in KSAO (31.00%), IKAP (30.77%) and CHAP (13.36%), while the cropland occupied by construction land was mainly concentrated in IKAP (33.57%), TCAO (16.07%) and KSAO (19.65%). From 2005 to 2010 (Figure 5b), the conversion of cropland into grassland and unused land was mainly distributed in KSAO (34.58%, 42.68%), TCAO (16.12%, 17.77%) and ASAO (12.19%, 14.89%). From 2010 to 2015 (Figure 5c), driven by the combined factors of returning cropland to grassland and urbanization, the scale of cropland loss in Xinjiang reached a peak. The conversion of cropland into grassland mainly occurred in CHAP (17.93%), IKAP (15.65%) and ATAO (13.31%) in northern Xinjiang, while the occupation of cropland by urbanization was mainly concentrated in CHAP (20.47%), UMQC (14.88%), KSAO (14.57%)

and IKAP. From 2015 to 2020 (Figure 5d), under China's ecological civilization system, Xinjiang's cropland beyond the capacity of water resources was returned to grassland or turned into unused land. Among them, returning cropland to grassland was mainly concentrated in KSAO (17.12%), ASAO (15.10%) and CHAP (14.74%), while the conversion of cropland into unused land mainly occurred in ASAO (19.92%) and KSAO (19.77%) in southern Xinjiang. Finally, the occupation of cropland by urbanization was distributed in all prefectures, especially in IKAP (15.28%), ASAO (13.95%), TCAO (13.10%) and CHAP (12.44%).

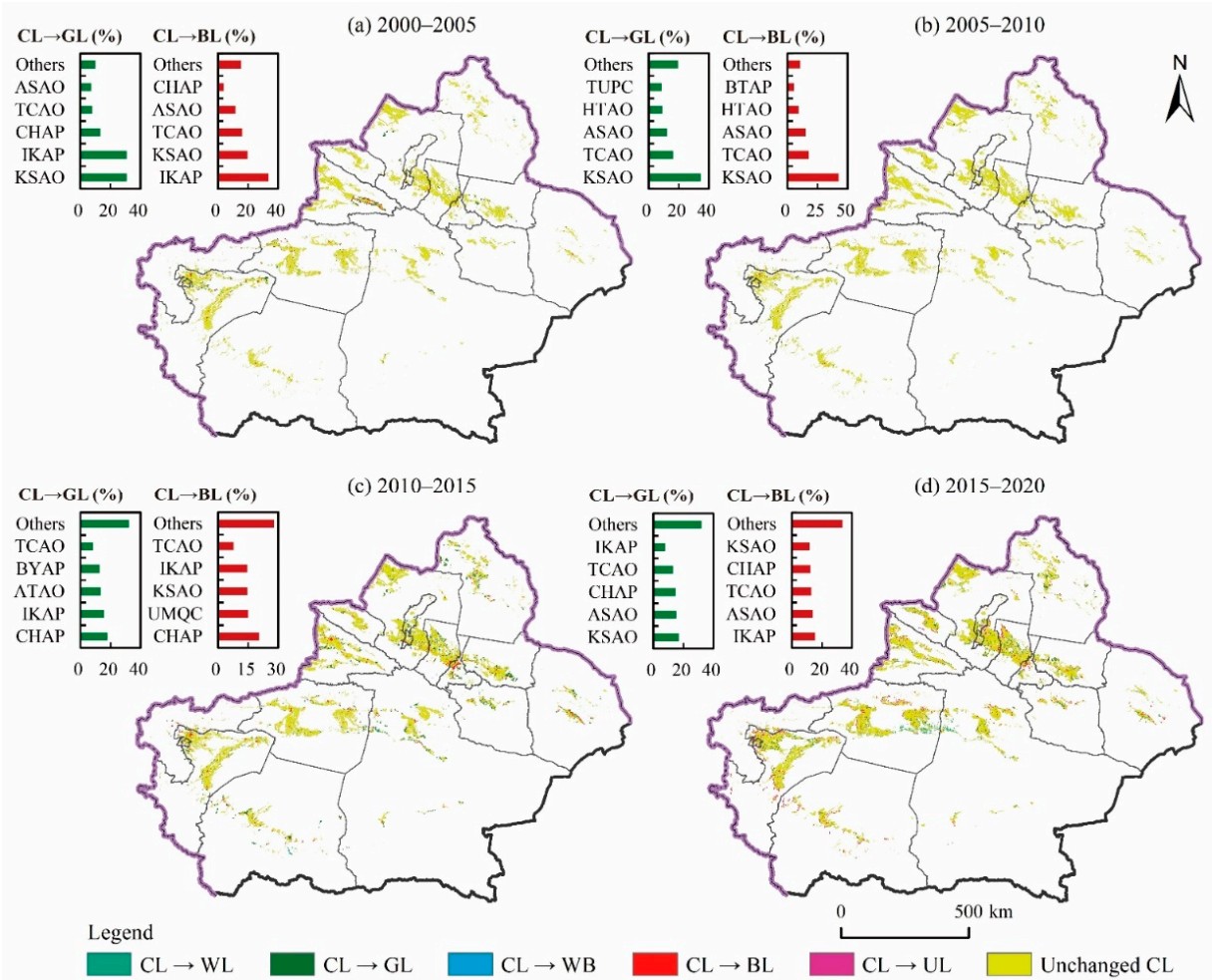

**Figure 5.** Spatial distribution of lost cropland in Xinjiang: (**a**) during 2000–2005; (**b**) during 2005–2010; (**c**) during 2010–2015; and (**d**) during 2015–2020.

*3.3. Response of the TESV to Cropland Use Change*

3.3.1. Response of the TESV to Cropland Use Change in Xinjiang Scale

Table 8 and Figure 6 report the changes in the TESV in response to cropland use changes in Xinjiang during the study period. Overall, these changes have significantly impacted the changes in the TESV; in this regard, the values of change in the total TESV were −1066.65, −588.08, −1015.75 and 51.98 million USD during 2000–2005, 2005–2010, 2010–2015 and 2015–2020, respectively. The impact of the conversion of cropland into other land types on the TESV was mainly positive, and the values of change in the TESV during the four periods were 23.86, 1.21, 19.33 and 29.80 million USD, respectively. Conversely, the impact of the conversion of other land types into cropland on the TESV was mainly negative, and the values of change in the TESV during the four periods were −1090.51, −589.29, −1035.08 and 22.18 million USD, respectively. Ultimately, the negative effect size

of the TESV generated by cropland expansion was much higher than the positive effect size of the TESV generated by cropland loss.

**Table 8.** Changes in the TESV caused by the conversion of cropland in Xinjiang during 2000–2020.

| Conversion Type | Period | Change in TESV (Million USD) | | | | |
|---|---|---|---|---|---|---|
| | | Provision Services | Regulatory Services | Support Services | Cultural Services | Total TESV |
| CL is converted to other land use types | 2000–2005 | −11.98 | 22.02 | 5.27 | 8.55 | 23.86 |
| | 2005–2010 | −3.21 | 3.04 | −0.36 | 1.74 | 1.21 |
| | 2010–2015 | −52.21 | 33.86 | 9.50 | 28.18 | 19.33 |
| | 2015–2020 | −15.44 | 33.79 | 0.11 | 11.34 | 29.80 |
| Other land use types are converted to CL | 2000–2005 | 216.15 | −637.46 | −442.83 | −226.37 | −1090.51 |
| | 2005–2010 | 136.76 | −338.14 | −254.32 | −133.59 | −589.29 |
| | 2010–2015 | 243.30 | −622.63 | −413.48 | −242.27 | −1035.08 |
| | 2015–2020 | 26.11 | −2.94 | 11.64 | −12.63 | 22.18 |

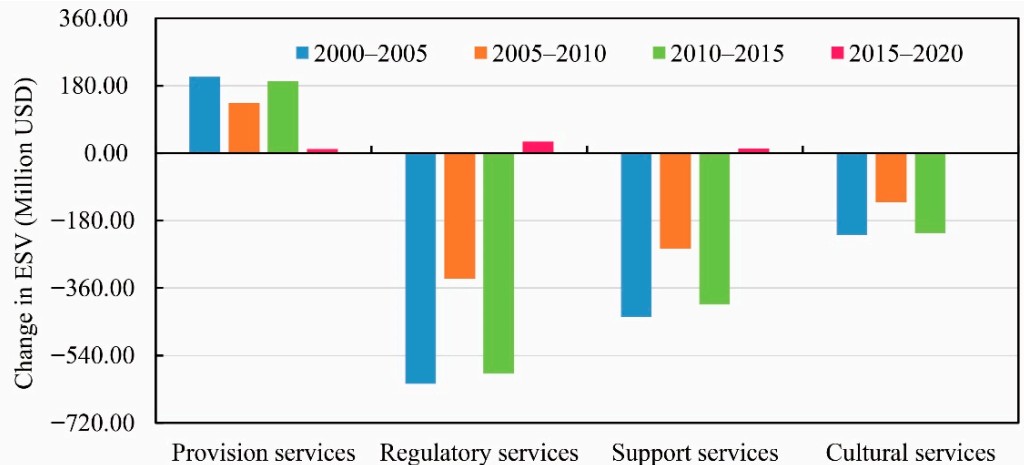

**Figure 6.** The impact of cropland conversion on the net value change of different ecosystem service types in Xinjiang during 2000–2020.

Figure 6 shows that the effect of cropland use conversion on the TESV changes in four different types of ecosystem services is discernibly different in Xinjiang. Under the comprehensive impact of cropland use conversion, the cumulative net value of ecosystem provision services increased by 539.49 million USD, while the cumulative net value of ecosystem regulation services, support services and cultural services decreased by 1508.47, 1084.47 and 565.05 million USD, respectively, from 2000 to 2020. Although the dominance of the expansion of cropland use in Xinjiang has generally enhanced the supply services value of the ecosystem, it has significantly sacrificed the value of ecosystem regulation, support and cultural services, demonstrating a significant trade-off.

### 3.3.2. Response of the TESV to Cropland Use Change in Xinjiang's Prefectures Scale

Table 9 and Figure 7 report the changes in the TESV caused by cropland use conversion in various prefectures of Xinjiang during the study period. These changes demonstrated significant differences at the prefecture level in Xinjiang.

**Table 9.** Changes in the TESV caused by the conversion of cropland in different prefectures of Xinjiang during 2000–2020.

| Name of District | Change in TESV (Million USD) | | | | | | | |
|---|---|---|---|---|---|---|---|---|
| | CL Is Converted to Other Land Use Types | | | | Other Land Use Types Are Converted to CL | | | |
| | 2000–2005 | 2005–2010 | 2010–2015 | 2015–2020 | 2000–2005 | 2005–2010 | 2010–2015 | 2015–2020 |
| ASAO | 4.70 | −0.02 | −3.25 | 15.92 | −144.49 | −148.80 | −307.82 | −8.38 |
| KSAO | 4.11 | −1.73 | −8.46 | 8.68 | −80.49 | −122.34 | −226.15 | −8.09 |
| TCAO | −3.02 | 0.74 | −2.25 | 1.54 | −245.12 | −31.44 | −111.98 | 2.39 |
| CHAP | 3.81 | 2.21 | −10.63 | 0.24 | −86.63 | −61.55 | −102.33 | −4.86 |
| IKAP | 13.15 | 0.14 | −3.82 | −1.75 | −167.72 | −32.58 | −97.64 | −6.43 |
| BYAP | 0.72 | 0.19 | 18.50 | 3.87 | −52.35 | −70.51 | −102.05 | −4.10 |
| ATAO | 1.73 | 0.51 | 11.64 | 4.66 | −114.44 | −34.41 | 7.10 | 50.76 |
| HTAO | −0.98 | −0.53 | 40.29 | 2.73 | −24.43 | −23.73 | −23.81 | 4.06 |
| BTAP | 0.83 | −0.43 | −2.09 | −1.65 | −72.46 | −20.52 | −31.19 | −1.31 |
| HAMC | −0.74 | 0.34 | −5.64 | −0.29 | −19.02 | −16.36 | −6.51 | 0.24 |
| KKAP | 0.68 | −0.13 | 1.46 | 1.62 | −2.29 | 0.03 | −15.33 | −1.66 |
| TUPC | 0.11 | 0.50 | 1.33 | −0.12 | −1.44 | 2.19 | −0.78 | 0.33 |
| KRMC | −0.47 | 0.00 | 2.48 | 0.09 | −69.16 | −16.18 | −11.09 | −0.52 |
| UMQC | −0.75 | −0.59 | −20.21 | −5.72 | −10.43 | −13.10 | −5.52 | −0.26 |

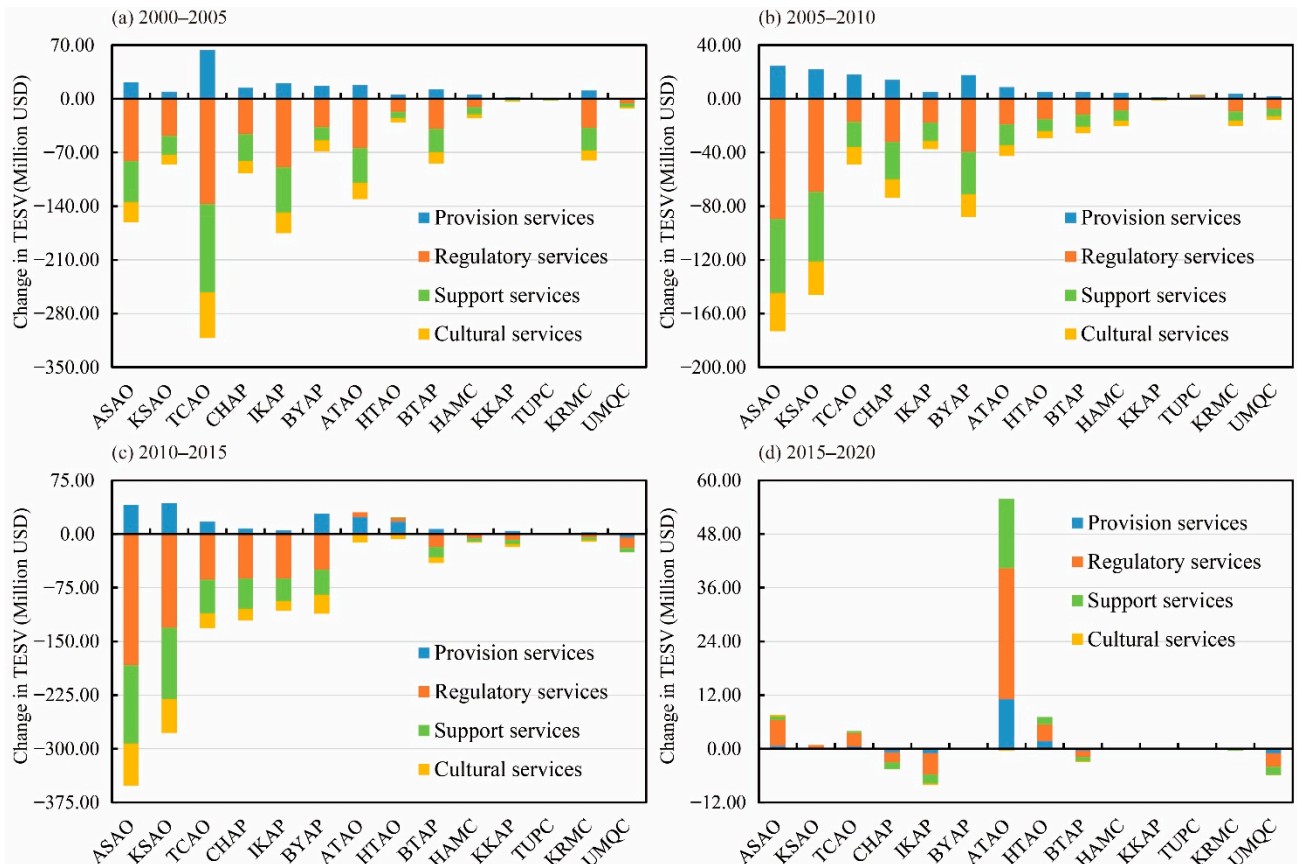

**Figure 7.** Net changes in four types of the TESV caused by the cropland conversion in different prefectures of Xinjiang during 2000–2020.

Regarding the impact of the conversion of cropland into other land types on the changes in the TESV (Table 9), during 2000–2005, nine prefectures (IKAP, etc.) showed positive effects, while five prefectures (TCAO, etc.) displayed negative effects. During 2005–2010, eight prefectures (CHAP, etc.) exhibited positive effects, while six prefectures (KSAO, etc.) showed negative effects. During 2010–2015, eight prefectures (UMQC, etc.)

demonstrated positive effects, while six prefectures (HTAO, etc.) revealed negative effects. During 2015–2020, nine prefectures (ASAO, etc.) showed positive effects, while five prefectures (UMQC, etc.) presented negative effects.

Regarding the impact of the conversion of other land types into cropland on the changes in the TESV (Table 9), during 2000–2005, all prefectures in Xinjiang demonstrated negative effects, among which the scale of negative effects in TCAO was prominent. During 2005–2010, except for TUPC and KKAP, the other 12 prefectures showed negative effects. During 2010–2015, except for ATAO, the other thirteen prefectures displayed negative effects. During 2015–2020, nine prefectures (ASAO, etc.) demonstrated negative effects, while five prefectures (ATAO, etc.) presented positive effects.

Under the comprehensive impact of cropland use conversion, net changes in the four types of ecosystem services in Xinjiang also differed during 2000–2020 (Table 9 and Figure 7). Regarding the change in the net value of ecosystem supply services, except for UMQC (−2.48 million USD), all other prefectures demonstrated positive effects. The top five prefectures with positive effects were TCAO (100.22 million USD), ASAO (86.61 million USD), KSAO (74.89 million USD), BYAP (62.82 million USD) and ATAO (61.49 million USD). With respect to ecosystem regulation services, except for TUPC (1.29 million USD), the other prefectures presented negative effects; the top five prefectures with negative effects were ASAO (−347.52 million USD), KSAO (−247.54 million USD), TCAO (−215.15 million USD), IKAP (−173.30 million USD) and CHAP (−141.94 million USD). Concerning the change in the net value of ecosystem support services, all prefectures showed negative effect; the top five prefectures with negative effects were ASAO (−218.97 million USD), TCAO (−180.79 million USD), KSAO (−177.64 million USD), CHAP (−107.51 million USD) and IKAP (−107.08 million USD). In terms of the change in the net value of ecosystem cultural services, all prefectures showed negative effects; the top five prefectures with negative effects were ASAO (−112.26 million USD), TCAO (−93.42 million USD), KSAO (−84.19 million USD), BYAP (−57.77 million USD) and IKAP (−46.68 million USD).

## 4. Discussion

### 4.1. Key Characteristics and Causes of Changes in Cropland Use in Xinjiang

Our research demonstrates that the cropland area in Xinjiang increased from 6.5682 million ha to 8.9874 million ha during 2000–2020, with an average annual expansion rate of 1.84%. The use of cropland has demonstrated significant expansion characteristics, and this supports scholars' conclusion that the Xinjiang oasis area is a hotspot for the northward migration of cropland space in China [21–23].

In terms of temporal changes in the scale of cropland use, previous studies have ascertained that cropland expansion in Xinjiang clearly displays a time-phased characteristic, and the expansion speed was the fastest during 2000–2010 [30,33]. Our research also confirms this conclusion. However, when we updated the research timeline to 2020, we were surprised to find that the scale of cropland use in Xinjiang had mostly remained stable since 2015, and the disorderly expansion observed during the early stage had been effectively curbed. A possible explanation for this phenomenon is that the Chinese government was vigorously implementing the ecological civilization strategy and development concept during 2015–2020. As of 2015, agricultural water consumption accounted for 95% of Xinjiang's total water consumption (57.72 billion $m^3$), and Xinjiang's total water consumption has far exceeded Xinjiang's 2020 total water use control target (52.67 billion $m^3$) determined in "Opinions on Implementing the Strictest Water Resources Management System" issued by the State Council of China [30]. Under its influence, the effects of the rigid constraints regarding water resources on the expansion of cropland in Xinjiang have become prominent [51], and the utilization of cropland in Xinjiang has also shifted from the disorderly expansion in the past to the stage of land withdrawal and water reduction.

Regarding the spatial characteristics of the scale of cropland use, we ascertained that the cropland use in all the prefectures in Xinjiang demonstrated a trend of expansion during the study period; however, the spatial centralized distribution characteristics of cropland

expansion were more significant. Nearly 71% of the cropland expansion in Xinjiang occurred mainly in ASAO, KSAO and BYAP in southern Xinjiang, as well as in TCAO and ATAO in northern Xinjiang. This finding aligns with related research conclusions drawn by scholars on the scale of the Xinjiang oasis area [30]. This may be the result of the cumulative effect of natural factors such as topography, temperature and precipitation and social-economic factors such as population size, urbanization level, non-agricultural industry development, irrigation conditions, interest inducement and agricultural policies in various prefectures in Xinjiang [30,31,33,34,52].

With respect to cropland use conversion, our research indicates that although Xinjiang is a hotspot for newly added cropland in China, there still exists a fierce mutual conversion between cropland and other land use types in Xinjiang, and the expansion of new cropland is accompanied by the loss of original cropland. Under China's Grain for Green policy and the impact of urbanization, grassland and construction land are the two main destinations of the lost cropland in Xinjiang. However, encroaching on grassland and reclaiming unused land are the two main modes of cropland expansion in Xinjiang. Additionally, we also found two strange phenomena. First, the implementation of the national Grain for Green project in 2000 was an important reason for the loss of cropland in China [21,22]. Although Xinjiang is one of the key areas for the implementation of the aforementioned project in China, our study reports that the contribution rate of grassland to the newly added cropland in Xinjiang accounted for 89.43%, 87.25%, 78.70% and 30.08% during the four periods from 2000 to 2020, respectively. This indicates that the expansion of cropland in Xinjiang mainly occurs at the cost of sacrificing grassland areas, and the intensity of cropland reclamation is far greater than the implementation intensity of the project to return cropland into grassland in the same period. Second, in the process of urbanization in Xinjiang, along with the encroachment of urban space on cropland areas, cropland is also constantly occupying ecological land to make up for the losses and expand the scale of cropland use. Compared with the current situation of urban expansion and cropland reduction in the process of urbanization in eastern and central regions of China [20,21], Xinjiang showed instead a double increase in urban expansion and cropland expansion, exerting tremendous pressure on the already fragile ecological environment in Xinjiang. This phenomenon can be attributed to Xinjiang's lagging urbanization process and the fact that its economic foundation heavily relies on agriculture. From 2000 to 2020, the population urbanization rate of Xinjiang increased from 33.75% to 56.53%, and the proportion of the primary industry in the industrial structure gradually decreased from 20.1% to 14.4% [48]. Xinjiang's urbanization level and industrial structure have shown a progressive trend. However, in 2020, Xinjiang's urbanization level was low and the proportion of agriculture in the national economy was high compared with China's population urbanization rate (63.89%) and the proportion of the primary industry (7.7%); moreover, the channels for rural labor to transfer to non-agricultural employment are narrow. Consequently, a large number of rural laborers relying on cropland resources for a career resort to the traditional planting industry to maintain their livelihood [11], thus forming a vicious cycle between population, cropland and agriculture.

### 4.2. Effect of Cropland Conversion on the Regional TESV in Xinjiang

The development and utilization of cropland significantly impact the sustainable development of terrestrial ecosystems. Studies have reported that global cropland expansion has resulted in a significant negative effect on the TESV and that cropland expansion on tropical forests was the main reason for global decrease in the TESV, especially in South America, Africa and Asia. Meanwhile, the conversion of grassland into cropland has played an important role in the reduction of the TESV in Oceania, Africa and Asia [8]. Our study indicates that the frequent conversion between cropland and other types of land in Xinjiang led to TESV changes amounting to −1066.65, −588.08 and −1015.75 million USD during three periods from 2000 to 2015, respectively, demonstrating significant negative effects (Table 8). This conclusion is consistent with the findings presented in other studies

based on typical case areas in Xinjiang [36,39]. On the one hand, in the equivalent factor table of ESV (Table 1), the equivalent factor of service value of grassland, woodland and wetland ecosystems is higher than that of cropland ecosystems, which has a greater impact on the changes in the TESV caused by the conversion of cropland. On the other hand, the expansion of cropland in Xinjiang took a substantial amount of ecological land from 2000 to 2015 (Table 6), and this had a significant negative impact on the TESV. From 2015 to 2020, the scale of cultivated land in Xinjiang remained stable on the whole due to the policy constraints of the "three red lines of water resources" [51]. The main source of new cropland became unused land (the contribution rate was 53.62%). Overloaded cropland was returned to ecological land under the constraints on water resources (the contribution rate was 40.38%). Consequently, these factors turned the changes in the TESV caused by cropland use conversion from negative to positive in Xinjiang at this stage, with a net added value of 51.98 million USD.

Judging from the response of the values of changes in the four types of ecosystem services to the cropland conversion, our research ascertained that the cumulative net value of ecosystem provision services increased by 539.49 million USD in Xinjiang from 2000 to 2020, while the cumulative value of ecosystem regulation services, support services and cultural services decreased by 1508.47, 1084.47 and 565.05 million USD, respectively. This demonstrates that although the dominance of cropland expansion in Xinjiang has significantly enhanced the supply services of the terrestrial ecosystem, it has brought greater losses to ecosystem regulation services, support services and cultural services and has ultimately caused the overall loss in the TESV. This finding has been supported by case studies in areas with extensive human agricultural activities [35,36,39,43,44].

Xinjiang is the main area of China's arid region, with an arid climate, water shortages and a fragile ecological environment [26]. It is geographically far from China's main grain-producing areas, and cropland use plays an important role in Xinjiang's grain self-sufficiency. Additionally, Xinjiang's lagging urbanization level and its economic foundation that heavily relies on agriculture position the use of cropland as a stabilizer for farmers' employment and income increases [11]. Therefore, Xinjiang's social stability and economic development require the maintenance of a certain scale of cropland use. However, the use of cropland in Xinjiang has displayed a high intensity of disorderly expansion, thereby making the region a hotspot of newly added cropland in China during 2000–2015. This not only sacrificed a large amount of natural ecological land (mainly grassland and unused land) but also caused a significant decrease in the TESV in Xinjiang. In fact, oasisization and desertification are two basic geographic processes that mutually affect each other in arid regions. The process of oasisization is mainly manifested in cropland expansion; however, excessive cropland expansion inevitably breaks the dynamic balance between oasisization and desertification and then accelerates the process of desertification [53]. Grassland and desert ecosystems in arid areas play crucial roles in water conservation, wind prevention, sand fixation, soil and water conservation, carbon fixation, oxygen release and biodiversity conservation [54–57]. The excessive encroachment and destruction of these ecosystems by cropland can significantly improve the provision services of the regional ecosystem in the short term but will cause irreversible heavy losses to the regulation and support services of the regional ecosystem in the long run.

Therefore, we need to rethink the sustainability of the dynamic balance of total cropland formed by the spatial transfer of cropland in China in recent decades. Not only does this phenomenon conceal the spatial mismatch between cropland quality and food productivity [28], as well as the severe negative impacts on irrigation water use, fertilizer use and natural habitats in newly developed agricultural areas [23], but it also has the potential to cause a substantial loss in the TESV in newly developed agricultural areas and to seriously threaten the stability and security of its ecosystem—as our research concludes. Moreover, how to coordinate the contradiction between Xinjiang's agricultural development and ecological security and how to control the appropriateness of cropland quantity are urgent

issues that must be addressed to ensure sustainable management of natural resources in Xinjiang.

*4.3. Limitation and Future Work*

Although we used high-resolution, highly timely and finely classified remote sensing monitoring data of land use and revised the ESV equivalent factor table based on the actual food production and price levels in the study area, our study still has some shortcomings. First, the equivalent factor method we adopted is a static evaluation method [46], and it does not consider the differences in ecosystem quality and monetary inflation over different years, and so there is a certain uncertainty. In future research, it is necessary to further revise the equivalent factor table of ESV based on the temporal and spatial changes of regional biomass and consider the inflation of currencies over different years. The purpose is to establish a relatively objective and comprehensive dynamic assessment method for the TESV, so as to minimize the uncertainty of the assessment results. Second, this study quantitatively reports the impact of cropland use conversion in Xinjiang on the values of changes in regional terrestrial ecosystem provision services, regulation services, support services and cultural services; however, it does not quantitatively analyze the trade-offs and synergies between these services caused by cropland conversion. Hence, follow-up research should consider introducing the evaluation model of the relationship between ecosystem service functions, and the ecosystem service function supply and demand evaluation model, to conduct a systematic evaluation and try to address the issue regarding the appropriate cropland scale required for realizing the coordinated guarantee of food security and ecological security in Xinjiang from the perspective of integrated management of ecosystem services.

## 5. Conclusions

This study focused on Xinjiang as China's hotspot for newly added cropland and employs five phases (2000, 2005, 2010, 2015 and 2020) of high-resolution and finely classified remote sensing monitoring data of land use. The impacts of changes in cropland use on the TESV in Xinjiang from 2000 to 2020 were evaluated based on the benefit transfer method and the improved ecosystem services unit value table. The main conclusions are as follows:

1.  In terms of change in the cropland amount, the cropland area in Xinjiang grew from 6.5682 million ha in 2000 to 8.9874 million ha in 2020, with an average annual expansion rate of 1.84%, undergoing four stages: rapid expansion (2000–2005), steady expansion (2005–2010), rapid expansion (2010–2015) and relative stability (2015–2020). Although the cropland quantity of all prefectures in Xinjiang displayed a trend of expansion, nearly 71% of the cropland expansion mainly occurred in ASAO, KSAO and BYAP in southern Xinjiang, as well as in TCAO and ATAO in northern Xinjiang, and the cropland expansion demonstrated a significant spatial concentration distribution feature.

2.  In terms of cropland use conversion, although Xinjiang is a hotspot for newly added cropland in China, a sharp mutual conversion trend was observed between cultivating cropland and other types of land use in Xinjiang from 2000 to 2020. In particular, grassland and unused land are the main sources of cropland expansion in Xinjiang, while lost cropland in Xinjiang is mainly returned to grassland and occupied by construction space. The mutual conversion between cropland and other types of land use is relatively weak.

3.  In terms of the TESV response, the values of change in the total TESV caused by cropland conversion in Xinjiang were −1066.65, −588.08, −1015.75 and 51.98 million USD during the four periods from 2000 to 2020, respectively. Although the intensive expansion of cropland and the conversion of cropland mainly at the expense of ecological land in Xinjiang have significantly enhanced the provision services of the terrestrial ecosystem (539.49 million USD), regulation services (−1508.47 million USD), support services (−1084.47 million USD) and cultural services (−565.05 million USD)

experienced greater losses. Ultimately, how to coordinate the contradiction between Xinjiang's agricultural development and ecological security, as well as how to control the appropriateness of cropland amount, are urgent issues that must be addressed to ensure sustainable management of natural resources in Xinjiang.

**Author Contributions:** Conceptualization—T.C. and X.Z.; methodology—T.C.; software—L.F. and J.H.; formal analysis—T.C.; data curation—X.L.; writing (original draft preparation)—T.C., X.L. and X.Z.; writing (review and editing)—T.C., X.L., L.F., J.H. and X.Z.; visualization—L.F. and J.H.; supervision—X.Z.; funding acquisition—X.Z. All authors have read and agreed to the published version of the manuscript.

**Funding:** This research was funded by the Third Xinjiang Scientific Expedition Program (grant number 2021xjkk0900) and the Comprehensive Geological Survey and Evaluation of Resource and Environment Carrying Capacity of Key Areas Along 315 National Highway in Southern Xinjiang (grant number ZD20220231).

**Institutional Review Board Statement:** Not applicable.

**Informed Consent Statement:** Not applicable.

**Data Availability Statement:** All data used for the study appear in the Data Sources section of the submitted article.

**Acknowledgments:** The authors acknowledge all colleagues and friends who voluntarily reviewed the translation of the survey and study manuscript.

**Conflicts of Interest:** The authors declare no conflict of interest.

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
