# Peer review of "The Impact of Cropland Use Changes on Terrestrial Ecosystem Services Value in Newly Added Cropland Hotspots in China during 2000–2020"

_land, doi:10.3390/land11122294_

Round 1
Reviewer 1 Report
The shifts in ecosystems in China are of world importance. The scale of these shifts is so great that an analysis such as this is necessary to understand what the world outside of China is facing, either for good or bad in terms of ecosystem changes. This is an important study done at a massive scale and subsequently of major importance. It is high-quality research conducted on an essential topic. The authors should be congratulated for their scientific work and sharing their findings. We now know much more about ecosystem shifts in this massive region than would have otherwise been possible. The conversion of these shifts to TESV is useful but less important than the record of the ecosystem shifts themselves. I would publish.
Author Response
Response to Reviewers Comments
Land-2075849
The Impact of Cropland Use Changes on Terrestrial Ecosystem Services Value in Newly Added Cropland Hotspots in China During 2000–2020
8-Dec-2022
Thank you for your positive and constructive comments and suggestions on our manuscript. According to these recommendations, we have made careful modification in our manuscript. The reviewers’ suggestions are marked in blue, and our responses are marked in black. The detailed information can also be seen in our revised manuscript (with changes marked). We used the revision mode of the Word Software to modify the manuscript. The main corrections and the responds to the Reviewer’s comments are as follows:
Response to Reviewer 1
- The shifts in ecosystems in China are of world importance. The scale of these shifts is so great that an analysis such as this is necessary to understand what the world outside of China is facing, either for good or bad in terms of ecosystem changes. This is an important study done at a massive scale and subsequently of major importance. It is high-quality research conducted on an essential topic. The authors should be congratulated for their scientific work and sharing their findings. We now know much more about ecosystem shifts in this massive region than would have otherwise been possible. The conversion of these shifts to TESV is useful but less important than the record of the ecosystem shifts themselves. I would publish.
R: Thank you very much for your recognition of our work. In the future, we will continue to carry out in-depth research in this field and strive to make more excellent research results. Best wishes to you!
- English language and style are fine/minor spell check required.
R: Thanks for the suggestion. We have done a language retouching of all the content of our manuscript through the professional language service company. The revised details can be found in the revised manuscript with changes marked.

Reviewer 2 Report
The article reports ambitious research on the effects of cropland expansion on the value of environmental services in Xinjiang. It employs the benefit transfer method to measure effects and trends. It is worthy work that deserves publication.
I offer a few thoughts for authors' consideration.1. The benefit transfer method is so central that its essence deserves terse characterization when first mentioned in the introduction. While the citations for it are strong, readers cannot be expected to find them in order to have a clear picture of what followss.
2. On the equation of line 216 and its description in line 221, we are confronted with the ratio 1/7. This assumes power in the analysis although its source is not clear to me. Because of its power, more basis ofr its use would be helpful.
3. The importance of water and water policy is not acknowledged until the last sections. This is uncomfortable. Some presentation of hydrological data would seem important for consideration of both cropland development and environmental services tied to water flow, timing, quality, and accessibility.
4. A trivial point is the difficulty I had with the maps of cropland development. It was very hard to see the yellow used for cropland. As cropland is the key focus, perhaps it deserves a map color that readers like me can see.
I hope these comments are helpful. They are merely intended to improve an already-fine work.
Author Response
Response to Reviewers Comments
Land-2075849
The Impact of Cropland Use Changes on Terrestrial Ecosystem Services Value in Newly Added Cropland Hotspots in China During 2000–2020
8-Dec-2022
Thank you for your positive and constructive comments and suggestions on our manuscript. According to these recommendations, we have made careful modification in our manuscript. The reviewers’ suggestions are marked in blue, and our responses are marked in black. The detailed information can also be seen in our revised manuscript (with changes marked). We used the revision mode of the Word Software to modify the manuscript. The main corrections and the responds to the Reviewer’s comments are as follows:
Response to Reviewer 2
- The benefit transfer method is so central that its essence deserves terse characterization when first mentioned in the introduction. While the citations for it are strong, readers cannot be expected to find them in order to have a clear picture of what follows.
R: Thank you for your suggestion. According to your suggestion, we have briefly introduced the essence of the benefit transfer method in the introduction of our manuscript. The revised details can be found in Line 502-505 of page 3 in the revised manuscript with changes marked.
- On the equation of line 216 and its description in line 221, we are confronted with the ratio 1/7. This assumes power in the analysis although its source is not clear to me. Because of its power, more basis of its use would be helpful.
R: Thank you for your reminding. 1/7 represents the ratio of the economic value provided by the natural ecosystem without human input to the economic value of grain production services provided by the per unit cropland area. In other words, the economic value of one ecological service value equivalent factor is equal to 1/7 of the national average grain yield market value in that year. This ratio (1/7) comes from the research of Chinese scholars Xie et al.
In 1997, Costanza proposed the evaluation principles and methods of 17 ecosystem services in 16 biomes around the world, which made the estimation of ESV clear in a scientific sense. Based on Costanza's work foundation and its shortcomings and combined with China's ecological characteristics, Chinese scholar Xie et al. put forward the evaluation method of ecosystem service value and equivalent factor table suitable for China's national conditions. This method has been widely used in the research on the value of ecosystem services at different scales in China, and our research is one of them.
In consideration of your suggestion, we have specified the literature source of this ratio (1/7) in our manuscript. The revised details can be found in Line 1033-1035 of page 6 in the revised manuscript with changes marked.
3 The importance of water and water policy is not acknowledged until the last sections. This is uncomfortable. Some presentation of hydrological data would seem important for consideration of both cropland development and environmental services tied to water flow, timing, quality, and accessibility.
R: Thank you for your suggestion. We would like to answer your question from three aspects.
First, we have mentioned in sections 3.2.1 and 3.2.2 of our manuscript that the reason for the small scale of new cropland in Xinjiang from 2015 to 2020 is that this period was rigidly constrained by the three red lines of water resources management and control system.
Second, we have added some hydrological data to Section 4.1 of our manuscript to further support our discussion. The details can be found in Line 2109-2113 of page 16 in the revised manuscript with changes marked.
Third, the main focus of our paper is on the impact of cropland use change on the service value of terrestrial ecosystem. Although water resources are scarce in arid areas, and the stress effect of cropland expansion on water resources is also very important, this is not the question we need to focus on in this article. In the future, we will definitely answer the stress effect of cropland use change on water resources in newly added cropland hotspots in China in the next article.
4 A trivial point is the difficulty I had with the maps of cropland development. It was very hard to see the yellow used for cropland. As cropland is the key focus, perhaps it deserves a map color that readers like me can see.
R: Thank you for your suggestion. We have modified the color of the cropland layer in Figure 4 and Figure 5 according to your suggestion. These can be found in Line 1728 of page 11 and Line 1754 of page 12 in the revised manuscript with changes marked.
- English language and style are fine/minor spell check required
R: Thanks for the suggestion. We have done a language retouching of all the content of our manuscript through the professional language service company. The revised details can be found in the revised manuscript with changes marked.

Reviewer 3 Report
This article analyzes the changes in cropland use in the Xinjiang region of China. The author uses the conventional analysis methods of land use change and discusses cropland use changes from 2000 to 2020 with the help of spatial overlay, land use transfer matrix, etc. And on this basis, the impact of ecosystem service value brought about by cropland change is analyzed. Overall, the full text lacks innovation and scientific nature, more like a research report. There are too many similar research contents in the field of LUCC. The authors should find their scientific problems rather than simply repeating the work many people have done. Since the research framework is too mediocre, I can't even offer effective comments to help you improve the research content. It is suggested that the author reorganize the literature. I think that discovering and proposing valuable topics may be your primary goal at present.
Author Response
Response to Reviewers Comments
Land-2075849
The Impact of Cropland Use Changes on Terrestrial Ecosystem Services Value in Newly Added Cropland Hotspots in China During 2000–2020
8-Dec-2022
Thank you for your positive and constructive comments and suggestions on our manuscript. According to these recommendations, we have made careful modification in our manuscript. The reviewers’ suggestions are marked in blue, and our responses are marked in black. The detailed information can also be seen in our revised manuscript (with changes marked). We used the revision mode of the Word Software to modify the manuscript. The main corrections and the responds to the Reviewer’s comments are as follows:
Response to Reviewer 3
- This article analyzes the changes in cropland use in the Xinjiang region of China. The author uses the conventional analysis methods of land use change and discusses cropland use changes from 2000 to 2020 with the help of spatial overlay, land use transfer matrix, etc. And on this basis, the impact of ecosystem service value brought about by cropland change is analyzed. Overall, the full text lacks innovation and scientific nature, more like a research report. There are too many similar research contents in the field of LUCC. The authors should find their scientific problems rather than simply repeating the work many people have done. Since the research framework is too mediocre, I can't even offer effective comments to help you improve the research content. It is suggested that the author reorganize the literature. I think that discovering and proposing valuable topics may be your primary goal at present.
R: Thank you for your comments. We will give a sincere explanation of your doubts in the following two aspects.
On the one hand, our study is not a research report. We have specific scientific problems, reasonable research methods and clear research conclusions. Our study aims to answer the question of how the spatial transfer of cropland in China affects the ecosystem service value of newly added cropland hotspots.
On the other hand, although our research is in the field of LUCC, and the methods used are also some classic mature methods, our research perspective is different from the existing research. Specifically, the spatial transfer of China's cropland distribution has maintained the dynamic balance of the total amount of cropland in China recently. However, Xinjiang (the hotspot area of newly added cropland in China) is located in the inland northwest of China and constitutes a major part of China’s arid area, with an extremely fragile ecological environment. Therefore, some scholars began to worry about and pay attention to the sustainability and ecological environmental risks of this special phenomenon. In this regard, the existing literature mainly refers to the impact of the spatial transfer of China's cropland on the spatial mismatch between food production and cropland resources, the natural suitability of cropland, the quality gap of cropland, as well as wind erosion, irrigation water consumption, fertilizer use and natural habitat in newly developed areas of cropland. However, the impact of China's cropland spatial transfer on terrestrial ecosystem services value (TESV) in hotspots of newly added cropland has not been substantial study. Some scholars discussed the impact of LUCC changes on ecosystem service value based on different case areas in Xinjiang. However, cropland is the only type of land use considered in these studies, thus offering insufficient importance and pertinence. Additionally, the research on individual typical regions in Xinjiang cannot address the question regarding the overall impact of China's newly added cropland transfer to the Xinjiang oasis area on its TESV. In this context, we carried out this study. Although our research is not a major original innovation, we have found a valuable research problem and clearly recognized the shortcomings of existing scholars in this area. We have also made improvements based on these deficiencies, and obtained more complete and comprehensive conclusions. This is also worthy of affirmation and appreciation.
Finally, I sincerely thank you for your comments on our manuscript and look forward to your approval of our work.
- Moderate English changes required.
R: Thanks for the suggestion. We have done a language retouching of all the content of our manuscript through the professional language service company. The revised details can be found in the revised manuscript with changes marked.

Reviewer 4 Report
This is an interesting study on the impact of the changes in cropland use on terrestrial ecosystem service value in new cropland hotspots in China. The authors have done a good job of collecting a unique dataset from the year 2000 – 2020. The paper is generally well-written and structured. Sufficient information about the literature review was presented for readers to follow the present study rationale and methods and also the findings were quite informative. However, in my opinion, the paper has some shortcomings regarding some texts. Below, I have provided some remarks on the text, which are often vague.
Specific Comments:
P1. Line 18, "...has not yet been comprehensively reviewed and reported,..." This sentence sounds like your article is a review article and an original article. Could you change this sentence to "... has not been substantial study on..." This would quickly help your readers know that you are talking about an original study.
P6. Line 224, Could you change the values in Table 1 to two decimal places just like you did in Table 2? Uniformity in data presentation is very important in a scientific article, as it would help for a better understanding.
P9. Line 308, could you change the values in Table 5 to two decimal places too?
Author Response
Response to Reviewers Comments
Land-2075849
The Impact of Cropland Use Changes on Terrestrial Ecosystem Services Value in Newly Added Cropland Hotspots in China During 2000–2020
8-Dec-2022
Thank you for your positive and constructive comments and suggestions on our manuscript. According to these recommendations, we have made careful modification in our manuscript. The reviewers’ suggestions are marked in blue, and our responses are marked in black. The detailed information can also be seen in our revised manuscript (with changes marked). We used the revision mode of the Word Software to modify the manuscript. The main corrections and the responds to the Reviewer’s comments are as follows:
Response to Reviewer 4
- This is an interesting study on the impact of the changes in cropland use on terrestrial ecosystem service value in new cropland hotspots in China. The authors have done a good job of collecting a unique dataset from the year 2000–2020. The paper is generally well-written and structured. Sufficient information about the literature review was presented for readers to follow the present study rationale and methods and also the findings were quite informative.
R: Thank you very much for your recognition of our work. In the future, we will continue to carry out in-depth research in this field and strive to make more excellent research results. Best wishes to you!
- P1. Line 18, "...has not yet been comprehensively reviewed and reported,..." This sentence sounds like your article is a review article and an original article. Could you change this sentence to "... has not been substantial study on..." This would quickly help your readers know that you are talking about an original study.
R: Thanks for the suggestion. We have revised this sentence according to your suggestion. These can be found in Line 17-18 of page 1 in the revised manuscript with changes marked.
- P6. Line 224, Could you change the values in Table 1 to two decimal places just like you did in Table 2? Uniformity in data presentation is very important in a scientific article, as it would help for a better understanding.
R: Thanks for the suggestion. We have reserved 2 decimal places of the data in Table 1 according to your suggestion. These can be found in Line 1032-1033 of page 6 in the revised manuscript with changes marked.
- P9. Line 308, could you change the values in Table 5 to two decimal places too?
R: Thanks for the suggestion. We have reserved 2 decimal places of the data in Table 5 according to your suggestion. The decimal places of the corresponding data in the analysis text have also been modified. These can be found in Line 1196 of page 9, in Line 1191-1192 of page 9 and 1670-1671 of page 10 in the revised manuscript with changes marked.
- English language and style are fine/minor spell check required.
R: Thanks for the suggestion. We have done a language retouching of all the content of our manuscript through the professional language service company. The revised details can be found in the revised manuscript with changes marked.

Round 2
Reviewer 3 Report
I have no further comments.